# Selecting the Target Population for Screening of Hepatic Fibrosis in Primary Care Centers in Korea

**DOI:** 10.3390/jcm11061474

**Published:** 2022-03-08

**Authors:** Huiyul Park, Eileen L. Yoon, Mimi Kim, Seon Cho, Jung-Hwan Kim, Dae Won Jun, Eun-Hee Nah

**Affiliations:** 1Department of Family Medicine, Myoungji Hospital, Hanyang University College of Medicine, Goyangsi 11749, Korea; bliss153@hanmail.net; 2Department of Internal Medicine, Hanyang University College of Medicine, Seoul 04763, Korea; mseileen80@gmail.com; 3Department of Radiology, Hanyang University College of Medicine, Seoul 04763, Korea; bluefish010@naver.com; 4Department of Laboratory Medicine, Health Promotion Research Institute, Seoul 07572, Korea; dduddi3755@hanmail.net; 5Department of Family Medicine, Uijeongbu Eulji Medical Center, Eulji University School of Medicine, Uijeongbusi 11749, Korea; 12thrib@hanmail.net

**Keywords:** fatty liver, health check-up cohort, hepatic fibrosis, magnetic resonance elastography, metabolic abnormality

## Abstract

Screening strategies for hepatic fibrosis are heavily focused on patients with fatty liver on sonography in primary care centers. This study aimed to investigate the target population for screening significant hepatic fibrosis in primary care centers. This retrospective cross-sectional cohort study used data from 13 nationwide centers. A total of 5111 subjects who underwent both abdominal sonography and magnetic resonance elastography as part of their health check-up were included. Subjects with viral hepatitis and/or a history of significant alcohol consumption were excluded. Significant and advanced hepatic fibrosis was defined as ≥3.0 kPa and ≥3.6 kPa in the MRE test, respectively. The prevalence of significant and advanced hepatic fibrosis was 7.3% and 1.9%, respectively. Among the subjects with significant hepatic fibrosis, 41.3% did not have fatty liver. Hepatic fibrosis burden increased according to the number of metabolic risk abnormalities. Nearly 70% of subjects with significant hepatic fibrosis also had two or more metabolic risk abnormalities and/or diabetes. However, the prevalence of fibrosis did not differ between the groups with and without fatty liver. The presence of two or more metabolic risk abnormalities was an independent risk factor for significant hepatic fibrosis regardless of the fatty liver. Therefore, in the setting of primary care centers, screening for hepatic fibrosis would better be extended to subjects with metabolically unhealthy status beyond those with fatty liver.

## 1. Introduction

The prevalence of significant hepatic fibrosis in the general population is 5.1–9.5% [1,2]. Considering the direct relationship between significant hepatic fibrosis and the risk of liver-related mortality [3,4], screening strategy and defining high risk group for significant hepatic fibrosis among the average risk group are important in primary care centers. Moreover, subjects with significant hepatic fibrosis have a very high risk of other coexisting gastrointestinal diseases (e.g., diverticulosis), extrahepatic malignancy, and cardiovascular disease beyond liver disease [5,6]. Although routine check-up or various types of health check-up programs are performed extensively at primary care centers, there is no consensus on the target population’s need for further evaluation for significant fibrosis. thus, the screening strategies for hepatic fibrosis are heavily focused on patients with fatty liver on sonography or elevated liver enzyme in health check-up programs [7,8,9]. This approach may have been based on the two-hit hypothesis of steatosis and oxidative stress that leads to fibrosis [10]. To date, the presence of fatty liver and elevated liver enzymes have been considered to be central dogma in the development of hepatic fibrosis.

However, recent published data suggested that metabolic abnormality and diabetes are recognized as potential risk factors for hepatic fibrosis [11,12] and liver-related outcomes [13,14,15]. A recent meta-analysis showed that the presence of diabetes mellitus increases the risk of severe liver disease outcomes by more than two-fold [13]. Two longitudinal studies also found that the number of metabolic risk abnormalities is significantly related to the occurrence of liver cirrhosis or hepatocellular carcinoma (HCC) [14] as well as all-cause and liver-related mortality in patients with nonalcoholic fatty liver disease (NAFLD) [15]. Despite these significant associations between metabolic risk abnormalities and worse liver outcomes, screening strategies for the high-risk group for hepatic fibrosis in primary care centers are mainly focused on patients with fatty liver on sonography.

This study aimed to evaluate the appropriate population in need of screening for significant hepatic fibrosis among subjects with metabolic abnormality from a primary care center.

## 2. Materials and Methods

### 2.1. Study Design

This cross-sectional retrospective study consecutively selected subjects who underwent either voluntary or obligatory occupational health check-ups, including voluntary magnetic resonance elastography (MRE), at 13 health check-up centers in Korea. This study was approved by the Institutional Review Board of Hanyang University Medical Center (IRB No. HY-2021-04-001-001). The requirement for informed consent was waived because of the retrospective nature of the study, and the analysis used anonymized clinical data.

The Korea Association of Health Promotion is a public institution that has been running a health check-up program for individuals and companies since 1983. Health check-up programs include not only those provided by the Korean National Health Insurance Service (NHIS) but also programs covered by private insurance or private expenses. It has 16 branches of health check-up centers placed in 6 metropolitan cities and 7 cities in Korea [4]. Among the 16 health check-up centers, the 13 health check-up centers that have MRE facilities installed were selected for the current study. The same protocol for evaluating liver fibrosis by MRE was also used at the 13 health check-up centers.

### 2.2. Rationale for Abdominal Sonography and MRE as Health Check-Up

Abdominal sonography is among the most widely performed basic examinations during health check-ups in Korea. It can be performed either by patient preference or during obligatory medical examinations provided every one or two years by certain groups or companies under the Act of Employment. In contrast, MRE is not included in the routine health check-up program. Nevertheless, there are various types of specialized health check-up programs, including MRE, for patients who decide to pursue a more comprehensive check-up. All examinations, including abdominal sonography and MRE, were conducted within one day.

The Korean NHIS provides an abdominal sonography and alpha-fetoprotein (AFP) test free of charge twice a year as a separate program for patients with chronic liver disease (viral hepatitis B and C and cirrhosis) [16]. Hence, patients with known chronic liver disease rarely choose MREs at their own expense.

### 2.3. Inclusion and Exclusion Criteria

The exclusion criteria were: (1) history of viral hepatitis; (2) significant alcohol consumption (weekly alcohol consumption >210 g for men and >140 g for women) according to the questionnaire response; (3) no simultaneous sonography; and (4) missing values in the biochemical tests that are prerequisites for metabolic syndrome assessment except for basal serum insulin level. A total of 8545 people underwent a health check-up between 1 January 2017 and 30 May 2020. Among them, subjects (*n* = 6775) who underwent both MRE and abdominal sonography were included in this study. Subjects with risks of chronic liver disease (*n* = 1665), such as subjects with chronic liver disease on past medical history, positive in viral markers, and having significant alcohol intake (weekly alcohol consumption >210 g for men and >140 g for women), were excluded. Finally, 5111 subjects were included in the analysis.

### 2.4. Clinical Parameters of Participants

Routine questionnaires were obtained from every examinee during the health check-up. It included questions regarding alcohol consumption (regularity of alcohol intake, number of intakes per week or month, and number of bottles during each intake) and history of metabolic risk abnormalities (diagnosis of hypertension, diabetes, and dyslipidemia, and corresponding medications). Anthropometric measurements included waist circumference, blood pressure, height, weight, body mass index (BMI, weight/height^2^), total fat mass, and lean mass. Additionally, fasting serum glucose, hemoglobin A1c (HbA1c), total cholesterol, low-density lipoprotein cholesterol, high-density lipoprotein cholesterol (HDL), triglycerides, aspartate aminotransferase (AST), alanine aminotransferase (ALT), and γ-glutamyl transferase (GGT) levels were measured. The medical records of the subjects were reviewed.

### 2.5. Assessment of Fatty Liver and Hepatic Fibrosis Severity

The presence of fatty liver was evaluated by sonography. Severity was graded as normal, mild, moderate, or severe based on the degree of fat infiltration [17]. Liver echotexture, attenuation, and visualization of the intrahepatic vessel borders and/or the diaphragm were used as indices.

Liver stiffness was measured using MRE. All MRE examinations were performed on an MRE hardware (GE Healthcare, Waukesha, WI, USA) with a 1.5 T imaging system using a two-dimensional MRE protocol [18]. The cut-off values for severity of liver fibrosis were set at MRE values; ≥stage 2 (namely ≥F2 or significant fibrosis), ≥3.0 kPa; ≥stage 3 (namely ≥F3 or advanced fibrosis), ≥3.6 kPa; and stage 4 (namely F4 or cirrhosis), ≥4.7 kPa [19].

### 2.6. Definition of Abnormality

Metabolic risk abnormalities were defined as follows [20]: (1) central obesity, waist circumference ≥80 cm for women and ≥90 cm for men; (2) high blood pressure, blood pressure ≥ 130/85 mmHg, and/or taking hypertension medication; (3) high triglyceride, serum triglyceride ≥ 150 mg/dL; (4) low-HDL cholesterol, serum HDL cholesterol level < 50 mg/dL for women and <40 mg/dL for men, and/or dyslipidemia medication; and (5) prediabetes or diabetes, fasting glucose level ≥ 100 mg/dL, HbA1c ≥ 5.7%, and/or taking diabetes medication. A metabolically unhealthy status was defined as having two or more metabolic risk abnormalities and/or diabetes, while a metabolically healthy status was defined as having less than two metabolic risk abnormalities and diabetes. All subjects were divided into the following four groups according to the presence of fatty liver and their metabolic health status: (A) metabolically healthy and non-fatty liver (MH-NFL); (B) metabolically unhealthy and non-nonfatty liver (MU-NFL); (C) metabolically healthy and fatty liver (MH-FL); and (D) metabolically unhealthy and fatty liver (MU-FL) (Figure 1 and Table 1). Abnormal aminotransferase levels were defined as serum AST or ALT levels > 40 IU/L.

### 2.7. Statistical Analyses

Continuous and categorical variables are presented as mean ± SD and number (%), respectively. These variables were analyzed using either the chi-square test or Fisher’s exact test and Student’s independent *t*-test. Odds ratios (ORs) for significant fibrosis were evaluated using multivariate logistic regression. Age, sex, presence of fatty liver, abnormal aminotransferase levels, and metabolically unhealthy status were included as variables in the multivariate logistic regression. Statistical analyses were performed using SPSS version 26.0, for Windows (SPSS Inc., Chicago, IL, USA), and statistical significance was set at *p* < 0.05.

## 3. Results

### 3.1. Baseline Characteristics

A total of 5111 subjects were included in this study (Figure 1). The mean age of this health check-up cohort was 46.9 years (Table 1). Comorbidities, such as hypertension, diabetes, and metabolic syndrome (having ≥3 metabolic risk abnormalities), were also present in 28.9%, 8.3%, and 23.3% of the subjects, respectively. All subjects were divided into four groups according to the presence of fatty liver and metabolic health status (Figure 1 and Table 1). Forty-seven percent of those subjects were metabolically unhealthy regardless of the presence of fatty liver (Table 1). Furthermore, 28% of patients with non-fatty liver (742/2628) were metabolically unhealthy.

### 3.2. Prevalence of Hepatic Fibrosis According to Metabolic Risk Factor Type

Within the total of the 5111 patients, 372 (7.3%) had significant fibrosis (MRE score ≥ 3.0 kPa), and 95 (1.9%) had advanced fibrosis (MRE score ≥ 3.6 kPa) (Table 2). The prevalence of advanced hepatic fibrosis was two times higher in prediabetes or diabetes subjects compare to total population (1.9% vs. 3.9%). Prevalence of significant and advanced hepatic fibrosis was 10.4% and 2.6%, respectively, in case of central obesity. Overall prevalence of subjects with central obesity, high blood pressure or hypertension medication, high triglyceride level, low HDL or dyslipidemia medication, and prediabetes or diabetes in total subjects were 33.5, 22.6, 35.6, 18.9, and 33.4%, respectively.

### 3.3. Hepatic Fibrosis Burden Increased When Subjects Had Two or More Metabolic Risk Abnormalities or Diabetes

The hepatic fibrosis burden increased with an increase in the number of metabolic abnormalities. When subjects had two or more metabolic risk abnormalities or diabetes, liver stiffness values significantly increased regardless of fatty liver status (Figure 2A). However, the hepatic fibrosis burden did not differ between subjects with one metabolic risk and those without any metabolic risk abnormality. The prevalence of significant fibrosis also had a similar pattern to that of the liver stiffness values (Figure 2B).

In order to evaluate the independent risk factor for significant hepatic fibrosis, univariate and multivariate analyses were done (Table 3). Age, male sex, BMI, and the presence of fatty liver, hypertension, diabetes, abnormal aminotransferase, and all five components of metabolic risk abnormalities in metabolic syndrome were associated with a higher risk of significant fibrosis in univariate analysis (Table 3a). Metabolically unhealthy status was evaluated as significant predictors in not only univariate but also multivariate analysis (OR 1.76, 95% CI 1.37–2.26, *p* < 0.001) (Table 3b), In contrast, the presence of fatty liver was not an independent risk factor for significant fibrosis.

### 3.4. Metabolically Unhealthy Status Shared Considerable Hepatic Fibrosis Regardless of Fatty Liver

A total of 41.1% and 39.0% of those with significant and advanced fibrosis did not combine fatty liver (black and red color) (Figure 3A,B). A total 67.5% and 74.7% of subjects with significant fibrosis and advanced hepatic fibrosis were metabolically unhealthy (red and blue color). The proportion of subjects with only fatty liver among those with significant and advanced fibrosis (green color) were 9.7% and 8.4%, respectively. The proportion of subjects with only metabolically unhealthy status among those with significant and advanced fibrosis (red color) were 18.3% and 22.1%, respectively. The proportion of hepatic fibrosis in subjects with metabolically unhealthy status (67.5% in significant, 74.7% in advanced) seemed higher than those in subjects with fatty liver (58.8% in significant, 61.0% in advanced) (Figure 3C). Liver stiffness values were significantly higher in the following order: MU-FL (2.43 ± 0.58 kPa), MU-NFL (2.33 ± 0.64 kPa), MH-FL (2.27 ± 0.42 kPa), and MH-NFL (2.24 ± 0.44 kPa) (Table 4). The prevalence of significant fibrosis among metabolically unhealthy subjects was not significantly different between the groups with or without fatty liver (11% vs. 9.2%, *p* = 0.166). Similarly, the prevalence of significant fibrosis among metabolically healthy subjects did not differ between the groups with or without fatty liver (4.4% vs. 4.5%, *p* = 0.868). In contrast, the prevalence of significant fibrosis was significantly higher in metabolically unhealthy subjects than in metabolically healthy subjects for those with (4.4% vs. 11%, *p* < 0.001) and without fatty liver (4.5% vs. 9.2%, *p* < 0.001).

## 4. Discussion

To date, screening strategies for significant and advanced hepatic fibrosis in the general population have focused on the presence of fatty liver. This is based on the assumption that most hepatic fibrosis cases not related to viral hepatitis and/or alcoholic liver disease could be attributed to NAFLD. Moreover, the key driver of NAFLD pathophysiology is attributed to the central dogma of intrahepatic fat accumulation, hepatic inflammation, and hepatic fibrosis. However, results of the present study showed that 41% of subjects from health check-up centers had no evidence of fatty liver disease even with significant hepatic fibrosis. This means that the sensitivity of fatty liver to screen for significant hepatic fibrosis in the health check-up setting is only approximately 40%. Therefore, 60% of subjects with significant hepatic fibrosis may be missed if fatty liver status remained the only focus.

It is well known that additional metabolic risk abnormalities for fatty liver increase the risk of hepatic fibrosis. However, there are insufficient data on the effect of each metabolic risk abnormality on the progression of hepatic fibrosis. Furthermore, the threshold for the need for hepatic fibrosis screening based on the sequence of increasing number of metabolic risk abnormalities is not known. The results of this study clearly showed that the prevalence of significant hepatic fibrosis significantly increased when two or more metabolic risk abnormalities were considered. The presence of two or more metabolic risk abnormalities and/or diabetes was an independent predictor of significant hepatic fibrosis, regardless of the fatty liver status. Therefore, active screening for hepatic fibrosis is necessary when people have diabetes or two or more metabolic abnormalities regardless of the presence of fatty liver.

In the present study, the rate of significant fibrosis in the MU-NFL group (9.2%), which was previously considered the highest risk group, was not different from that of the MU-FL group (11%). Additionally, there was no difference in the prevalence of fibrosis between the MH-NFL and MH-FL groups. This supports the hypothesis that metabolically unhealthy status, the presence of two or more metabolic risk abnormalities and/or having diabetes, should be an additional indication for hepatic fibrosis screening beyond the presence of fatty liver. Additionally, it should be emphasized that 4.5% of subjects in the MH-NFL group with no viral hepatitis and/or alcohol consumption had significant fibrosis. Therefore, there is a possibility that veiled risk factors (e.g., past hepatitis B infection, genetic differences or single nucleotide polymorphisms, intestinal microbial composition, and sarcopenia) may play a role in increasing the fibrosis burden [21,22,23].

Recently, several studies suggested that non-alcoholic fatty pancreas disease (NAFPD) might be use as surrogate marker for pre-diabetics or metabolic dysfunction [24]. Further studies needed on clinical implication of co-existence of NAFPD and NAFLD.

Careful interpretation is required at several points. First, the sensitivity of sonography to detect the presence of fatty liver is largely dependent on the skill and expertise of the examiners. However, most subjects with fatty liver are diagnosed with the abdominal sonography in real-life settings. Examining routine MRI in the diagnosis of fatty liver is unrealistic. Therefore, it would be reasonable to analyze the sonographic data, and it could strengthen the generalizability of our study. Second, the presence of fatty liver could be affected by the process of burning out (burnt-out cirrhosis) or active lifestyle modification. In our study, it is insufficient to discuss the mechanism of hepatic fibrosis in subjects without fatty liver. However, we want to emphasize that there is a risk of missing subjects with significant fibrosis when the presence of fatty liver is the only focus in the screening process for fibrosis.

This study has several limitations. First, selection bias could be inherent because the study design was based on the MRE test. MRE was performed only in the population willing to pay for this additional test in the routine health check-up. Further, men were also predominant, and the age of the included subjects was younger than the median age of the general population. These patients could also be those of the population with fear of chronic liver disease and higher socioeconomic status. Nevertheless, the health check-up program could be considered cost-effective by most Koreans considering the huge number of MRE tests performed during the two-year study period. The subjects with evidence of viral hepatitis or alcoholic liver disease were excluded from the analysis. Additionally, Korea has a separate health check-up program that provides free-of-charge abdominal sonography and AFP test twice a year for patients with chronic hepatitis (viral hepatitis B, viral hepatitis C, and liver cirrhosis) as surveillance tests for HCC. Therefore, it is assumed that there is a low chance of including patients with a known risk of hepatic fibrosis in chronic liver disease. Second, liver fibrosis was evaluated using the MRE in the current study. Although liver biopsy is considered the gold standard for evaluating hepatic fibrosis, it is not feasible to perform liver biopsy during a health check-up for a population with a lower chance of chronic liver disease than that of a hospital cohort. Third, the cut-off values for MRE in the evaluation of significant fibrosis did not reach a consensus. The 3.0 kPa value was used as a cut-off for significant fibrosis based on data from a recent meta-analysis of the general population. To compensate for this limitation, a sensitivity analysis was performed with various cut-off values (3.2 kPa and 3.4 kPa) to estimate significant fibrosis, and the results were not significantly different between the groups. Fourth, the assessment of significant alcohol intake was entirely based on the questionnaire results, and the possibility of underreporting alcohol consumption could not be excluded. This could be one of the reasons for the non-negligible rate of significant fibrosis in groups without fatty liver regardless of metabolic health status. Moreover, important metabolic risk abnormalities, such as serum insulin level and CRP, could not be included in this analysis because they were not examined routinely in the health check-up program. Fifth, the prevalence of advanced fibrosis in this study was considerably small (only 1.9% of the cohort) such that all analyses in this study were performed with significant fibrosis (stage 2).

In conclusion, 41% of subjects with significant hepatic fibrosis in the health check-up cohort did not have fatty liver. The prevalence of significant fibrosis was significantly higher in subjects with two or more metabolic risk abnormalities and/or diabetes. The prevalence of significant fibrosis was not different between the two groups with or without fatty liver when compared within each group of the same metabolic status. Therefore, in the setting of primary care centers, screening for hepatic fibrosis would better be extended to subjects with metabolically unhealthy status beyond those with fatty liver.

## Figures and Tables

**Figure 1 jcm-11-01474-f001:**
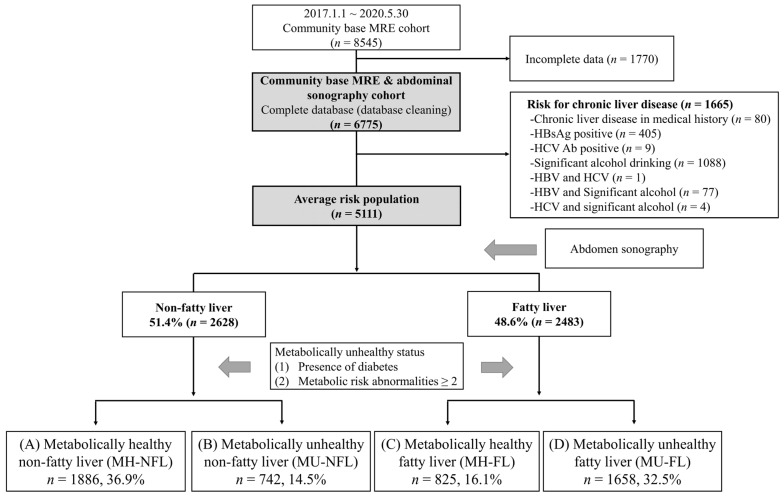
Flow diagram of study. Abbreviations: MH-NFL, metabolically healthy, non-fatty liver; MU-NFL, metabolically unhealthy, non-fatty liver; MH-FL, metabolically healthy, fatty liver; MU-FL, metabolically unhealthy, fatty liver; MRE, magnetic resonance elastography.

**Figure 2 jcm-11-01474-f002:**
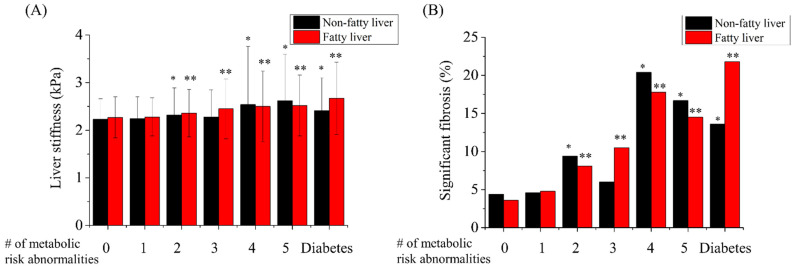
Hepatic fibrosis burden according to the number of metabolic risk abnormalities or the presence of diabetes. Liver stiffness (**A**) and prevalence of significant fibrosis (**B**) in the health check-up centers with and without fatty liver on sonography, according to the number of metabolic risk abnormalities or the presence of diabetes. * *p*-value was lower than 0.05 when compared with non-fatty liver subjects with 0 metabolic risk abnormality. ** *p*-value was lower than 0.05 when compared with fatty liver subjects with 0 metabolic risk abnormality.

**Figure 3 jcm-11-01474-f003:**
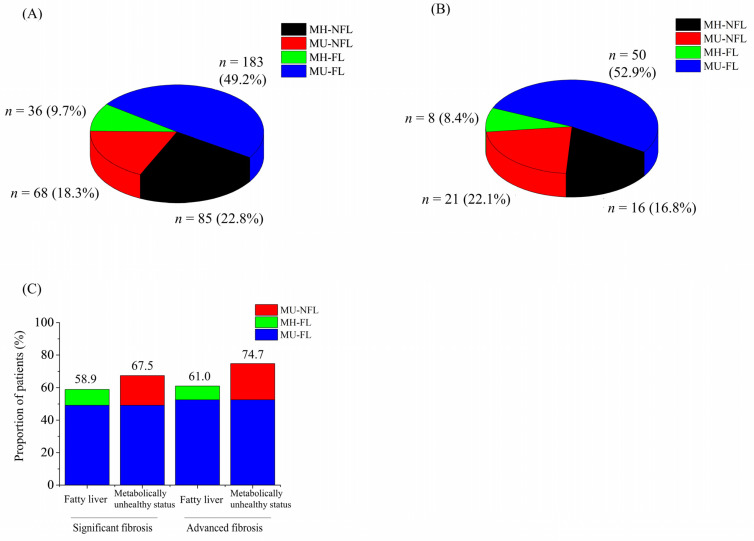
The hepatic fibrosis of the groups (A–D). Venn diagram representing the proportion of patients in groups A–D with (**A**) significant (*n* = 372) and (**B**) advanced fibrosis (*n* = 95). (**C**) The proportion of patients with fatty liver and metabolically unhealthy status with significant and advanced fibrosis. Abbreviations: MH-NFL, metabolically healthy, non-fatty liver; MU-NFL, metabolically unhealthy, non-fatty liver; MH-FL, metabolically healthy, fatty liver; MU-FL, metabolically healthy, fatty liver.

**Table 1 jcm-11-01474-t001:** Baseline characteristics according to presence of fatty liver and/or metabolic abnormality.

Characteristics	Total *n* = 5111	Non-Fatty Liver		Fatty Liver		*p*-Value					
		(A) MH-NFL *n* = 1886 (36.9%)	(B) MU-NFL *n* = 742 (14.5%)	(C) MH-FL *n* = 825 (16.1%)	(D) MU-FL *n* = 1658 (32.5%)	A vs. B	A vs. C	A vs. D	B vs. C	B vs. D	C vs. D
Age (years) †	46.9 ± 10.4	44.7 ± 10.5	50.6 ± 10.7	45.9 ± 9.3	48.2 ± 10	<0.001	0.004	<0.001	<0.001	<0.001	<0.001
Male/Female	4170/942 (81.6/18.4)	1349/537 (71.5/28.5)	592/150 (79.8/20.2)	730/95 (88.5/11.5)	1499/159 (90.4/9.6)	<0.001	<0.001	<0.001	<0.001	<0.001	0.136
Hypertension	1475 (28.9)	184 (9.8)	375 (50.5)	63 (7.6)	853 (51.4)	<0.001	0.078	<0.001	<0.001	0.681	<0.001
Diabetes	424 (8.3)	0 (0)	103 (13.9)	0 (0)	321 (19.4)	<0.001	-	<0.001	<0.001	0.001	<0.001
Number of metabolic risk abnormalities †	1.5 ± 1.3	0.4 ± 0.4	2.3 ± 0.7	0.6 ± 0.4	2.8 ± 0.8	<0.001	<0.001	<0.001	<0.001	<0.001	<0.001
Metabolic syndrome	1189 (23.3)	0 (0)	243 (32.7)	0 (0)	946 (57.1)	<0.001	-	<0.001	<0.001	<0.001	<0.001
BMI (kg/m^2^) †	24.8 ± 3.2	22.8 ± 2.3	24.9 ± 2.8	24.8 ± 2.4	27.1 ± 3	<0.001	<0.001	<0.001	0.79	<0.001	<0.001
Waist circumference (cm) †	85.5 ± 9	79.2 ± 7.2	86.3 ± 7.8	85.4 ± 6.3	92.3 ± 7.2	<0.001	<0.001	<0.001	0.013	<0.001	<0.001
Total fat mass (kg) †	18.5 ± 5.8	15.2 ± 4.1	18.7 ± 5.3	18.3 ± 4.2	22.4 ± 5.8	<0.001	<0.001	<0.001	0.085	<0.001	<0.001
Lean mass (kg) †	49.3 ± 8.8	46.1 ± 8.5	48.9 ± 8.6	49.7 ± 7.4	52.8 ± 8.5	<0.001	<0.001	<0.001	0.051	<0.001	<0.001
Lean mass/BW (%) †	68.7 ± 6.9	70.8 ± 6.8	68.4 ± 6.9	68.8 ± 6.1	66.3 ± 6.5	<0.001	<0.001	<0.001	0.227	<0.001	<0.001
SBP (mmHg) †	116 ± 13	111 ± 11	121 ± 14	113 ± 9	122 ± 13	<0.001	<0.001	<0.001	<0.001	0.192	<0.001
DBP (mmHg) †	75 ± 9	71 ± 7	78 ± 10	72 ± 7	78 ± 9	<0.001	0.001	<0.001	<0.001	0.09	<0.001
AST (IU/L) †	30 ± 18	25 ± 11	28 ± 15	29 ± 21	36 ± 22	<0.001	<0.001	<0.001	0.132	<0.001	<0.001
ALT (IU/L) †	32 ± 30	22 ± 21	27 ± 23	34 ± 36	45 ± 34	<0.001	<0.001	<0.001	<0.001	<0.001	<0.001
GGT (U/L) †	57 ± 88	37 ± 46	59 ± 70	55 ± 114	80 ± 110	<0.001	<0.001	<0.001	0.422	<0.001	<0.001
Triglyceride (mg/dL) †	145 ± 111	92 ± 51	162 ± 100	121 ± 69	211 ± 141	<0.001	<0.001	<0.001	<0.001	<0.001	<0.001
HDL (mg/dL) †	52 ± 12	59 ± 12	51 ± 12	52 ± 10	46 ± 10	<0.001	<0.001	<0.001	0.018	<0.001	<0.001
Glucose (mg/dL) †	99 ± 21	90 ± 8	105 ± 23	92 ± 8	109 ± 27	<0.001	<0.001	<0.001	<0.001	0.001	<0.001
HbA1c (%) †	5.7 ± 0.7	5.4 ± 0.3	5.9 ± 0.9	5.6 ± 0.3	6.1 ± 0.9	<0.001	<0.001	<0.001	<0.001	<0.001	<0.001

Data are expressed as number (percent). † Data are shown as mean ± standard deviation. Abbreviations: MH-NFL, metabolically healthy, non-fatty liver; MU-NFL, metabolically unhealthy, non-fatty liver; MH-FL, metabolically healthy, fatty liver; MU-FL, metabolically unhealthy, fatty liver; BMI, body mass index; SBP, systolic blood pressure; DBP, diastolic blood pressure; AST, aspartate aminotransferase; ALT, alanine aminotransferase; GGT, γ-glutamyl transferase; HDL, high-density lipoprotein; HbA1c, hemoglobin A1c.

**Table 2 jcm-11-01474-t002:** Prevalence of significant and advanced hepatic fibrosis of subjects with each component of metabolic risk abnormalities.

	Significant Fibrosis *	Advanced Fibrosis *
Total subjects *n* = 5111	372/5111 (7.3)	95/5111 (1.9)
Subjects with metabolic risk abnormality		
Central obesity *n* = 1812 (35.5)	189/1812 (10.4)	48/1812 (2.6)
High blood pressure or hypertension medication *n* = 1156 (22.6)	108/1156 (9.3)	29/1156 (2.5)
High triglyceride *n* = 1811 (35.6)	170/1811 (9.4)	44/1811 (2.4)
Low HDL or dyslipidemia medication *n* = 967 (18.9)	87/967 (9.0)	26/967 (2.7)
Prediabetes or diabetes *n* = 1705 (33.4)	207/1705 (12.1)	67/1705 (3.9)

Data are expressed as number (percent). * Prevalence of significant or advanced hepatic fibrosis indicates the proportion of subjects with significant or advanced hepatic fibrosis among total subjects (*n* = 5111), subjects with central obesity (*n* = 1812), high blood pressure or hypertension medication (*n* = 1156), high triglyceride (*n* = 1811), low HDL or dyslipidemia medication (*n* = 967), or prediabetes or diabetes (*n* = 1705). Abbreviations: HDL, high-density lipoprotein.

**Table 3 jcm-11-01474-t003:** Univariate (a) and multivariate (b) risk factor analysis of significant hepatic fibrosis.

Variables	OR	CI (95%)	*p*-Value
(a) Univariate analysis			
Age	1.04	1.03–1.05	<0.001
Male sex	2.13	1.50–3.02	<0.001
BMI (kg/m^2^)	1.12	1.09–1.16	<0.001
Fatty liver (≥mild)	1.56	1.26–1.93	<0.001
Hypertension	1.69	1.36–2.11	<0.001
Metabolic risk abnormalities (≥2)	2.54	2.03–3.17	<0.001
Metabolic risk abnormalities (≥3)	2.37	1.91–2.95	<0.001
Diabetes	3.77	2.89–4.92	<0.001
Metabolically unhealthy status	2.50	1.99–3.12	<0.001
Abnormal aminotransferase	2.96	2.39–3.68	<0.001
Component of metabolic risk abnormality			
Central obesity	1.98	1.60–2.45	<0.001
High blood pressure or hypertension medication	1.44	1.14–1.82	0.002
High triglyceride	1.58	1.27–1.95	<0.001
Low HDL or dyslipidemia medication	1.33	1.04–1.72	0.023
Prediabetes or diabetes	2.71	2.19–3.36	<0.001
(b) Multivariate analysis			
Age	1.04	1.03–1.05	<0.001
Male sex	1.85	1.29–2.65	0.001
Fatty liver (≥mild)	0.86	0.67–1.10	0.256
Abnormal aminotransferase	2.80	2.21–3.55	<0.001
Metabolically unhealthy status	1.76	1.37–2.26	<0.001

Odds ratios for significant hepatic fibrosis in (a) univariate analysis were calculated using the chi-square test. Odds ratios of fatty liver or metabolically unhealthy status for significant fibrosis in (b) multivariate analysis were evaluated through the multi-variate logistic regression analysis. Abbreviations: BMI, body mass index; OR, odds ratio; CI, confidence interval; HDL, high-density lipoprotein.

**Table 4 jcm-11-01474-t004:** Liver stiffness and prevalence of significant fibrosis among four groups (a) and liver stiffness among four group with significant fibrosis.

	(A) MH-NFL	(B) MU-NFL	(C) MH-FL	(D) MU-FL	*p*-Value					
					A vs. B	A vs. C	A vs. D	B vs. C	B vs. D	C vs. D
Liver stiffness (kPa) †	2.24 ± 0.44	2.33 ± 0.64	2.27 ± 0.42	2.43 ± 0.58	<0.001	0.044	<0.001	0.05	<0.001	<0.001
Significant fibrosis	85 (4.5)	68 (9.2)	36 (4.4)	183 (11)	<0.001	0.868	<0.001	<0.001	0.166	<0.001

Data are expressed as number (percent). † Data are shown as mean ± standard deviation. Abbreviations: MH-NFL, metabolically healthy, non-fatty liver; MU-NFL, metabolically unhealthy, non-fatty liver; MH-FL, metabolically healthy, fatty liver; MU-FL, metabolically healthy, fatty liver.

## Data Availability

Data available on request due to privacy/ethical restrictions.

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
