# Peer review of "Selecting the Target Population for Screening of Hepatic Fibrosis in Primary Care Centers in Korea"

_jcm, 2022, doi:10.3390/jcm11061474_

Round 1

Reviewer 1 Report

The manuscript by Dr Park and Yoon et al describes assessment of hepatic fibrosis by magnetic resonance elastography (MRE) in patients from 13 health care centers in Korea. In addition to MRE, patients were screened by abdominal ultrasound for the presence of fatty liver, for metabolic markers by blood test and by routine questionnaires and physical examination. After excluding patients with viral hepatitis and a risk of liver disease, 5111 patients were studied and were assigned to 4 groups: (A) metabolically healthy-no fatty liver (MH-NFL), (B) metabolically unhealthy no fatty liver (MU-NFL), (C) metabolically healthy-fatty liver (MH-FL) and (D) metabolically unhealthy fatty liver (MU-FL). The results of the study are interesting. A total of 7.3% of patients had significant fibrosis and of these 41% were assigned to the MH-NFL and MU-NFL groups indicating that patients with fibrosis might be missed if screening was restricted to patients with fatty liver detected by abdominal ultrasound. The presence of >2 metabolic abnormalities was also an independent risk factor for significant fibrosis.

However, the number of patients with significant or advanced fibrosis is low. Only 1.9% of patients had advanced fibrosis and because of the low patient numbers only limited analysis of patients in this category was provided. The manuscript must be revised to clearly define the number of patients with significant and advanced fibrosis. This could include a new Table 2 showing the number of patients in Groups A-D with significant and advanced fibrosis similar to that shown in the Table section of Figure 4. The number of patients at each level of hepatic stiffness could also be included in Table 4. The M&M section in particular must be revised to eliminate repetitive text.

Comments and suggestions:

  1. Line 2. Consider revising the title to: “Selecting the target population for screening of hepatic fibrosis in primary care centers in Korea.”
  2. Lines 3, 21, 34, 42-43, 46, 58, 68, 78-81, 87-88. Review use of the terms “health check-up centers”, “health check-up programs”, “health check-up cohorts”, “health promotion centers” and “primary care centers”. Can this be simplified?
  3. Lines 6-15. Remove email addresses. There is potential confusion with multiple authors from addresses 2 and 4.
  4. Lines 28 and 29. Review use of the terms metabolic “abnormalities” and metabolic “risks” to see if a consistent descriptor can be used. The words “dysfunction”, “syndrome”, “unhealthy”, “risk abnormalities” and “risks/dysfunction” are all used at other places in the manuscript.
  5. Revise to include details of the 1.9% of patients with advanced fibrosis.
  6. Lines 21, 24, 82, 89, 91, 101, 105, 106, 126, 227, 270, 328 and Figure 1. Review use of “ultrasonography”, “sonography”, “sonographic” and “ultrasound” to simplify the text.
  7. Line 40. Revise to: “defining which patients in the target population need further diagnostic”.
  8. Line 44. Revise to: “on the target population who need further”.
  9. Line 50. Revise to: “Recent published data suggested”.
  10. Line 73. Check spelling of “Data”.
  11. Lines 65-81. Revise and shorten and consider combining Sections 2.1 and 2.2.
  12. Lines 91-94. Is this information needed?
  13. Lines 96-99. Remove duplication with Section 2.2.
  14. Line 106. Delete “in the MRE and abdominal ultrasound cohorts”.
  15. Figure 1. This Figure can be moved to the Results, Section 3.
  16. Figure 1. The N=5111 should be n=5111. The details in the RH box “Chronic liver disease” and “Risk for liver disease” are confusing and should be simplified. Can a single heading be used?
  17. Figure 1. Check the box “Non-fatty liver”. Should this be “No fatty liver”?
  18. Figure 1. The category “Metabolically unhealthy” applies to Groups B and D and could be placed centrally or included in the footnotes.
  19. Lines 161-164. Is repetitive with Section 2.4 and Figure 1.
  20. Table 1. Include the number of female patients in each group. There may be a simpler way to present the P-value data as it doesn’t add much to the Table. The total of the percentages in the row “Characteristics” equals 99.9% not 100%.
  21. Lines 167-169. States that the % of patients with significant and advanced fibrosis (7.3% and 1.9%) is presented in Table 1. Can this detail be included in a new Table 2 suggested above?
  22. Lines 180-181. States that the prevalence of significant and advanced fibrosis was 12.1% and 3.9% with pre-diabetes or diabetes but the data is not shown. Can this detail be included in a new Table 2 suggested above?
  23. Table 2. Revise heading to “significant hepatic fibrosis”. Do males have a risk factor for fibrosis?
  24. Lines 193-201. Footnotes, Table 2. Revise and shorten and refer to M&M section.
  25. Figure 2A and 2B. The description of the P-values is confusing.
  26. Table 3. Models 1, 2 and 3 are not discussed and seem repetitive.
  27. Figure 3. It’s not immediately clear how Figures 3A and 3B relate to 3C. It would be helpful if the colors were matched in Figures 3A and 3B with Figure 3C.
  28. Figure 3C. The vertical axis is labelled “Hepatic fibrosis (%)” but all patients have been selected for “hepatic fibrosis”. Similarly, the heading to Figure 3C does not make sense.
  29. Legend to Figure 3C. Does not mention advanced fibrosis.
  30. Lines 259-261 and Figure 4A. Given the large error bars it is hard to believe there are statistical differences between Groups A-D in the levels of liver stiffness. Can the levels of liver stiffness for the Group A-D patients with significant and advanced fibrosis be provided in a new Table 2?
  31. Figure 4, Table. Consider incorporating the data from the Table into a new Table 2. Figure 4A and 4B if still considered essential could be incorporated into Figure 3.
  32. Table 4. It would be useful to add the number of patients at each level of hepatic stiffness (>3 kPa – >4.2kPa) to help support the data.
  33. Reference 2. Check page numbers.
  34. Reference 4. Check volume and year.
  35. Reference 8. Check page numbers.
  36. Reference 10. Check volume and page numbers.
  37. Reference 13. Check volume.

Author Response

Dear Reviewers and editors in JCM

Thank you very much for your time and consideration of our manuscript. We appreciate the opportunity to respond to the comments by the reviewers and to improve our paper. Please find below our responses to reviewers’ comments.

If we summarized the main change in Figure and tables.

HbA1c data was added in table 1, according to the recommendation of reviewer 2.

New table 2 and table 5 were added according to the recommendation of reviewer 1.

New table 2 shows prevalence of significant and advanced hepatic fibrosis of subjects with each component of metabolic risk abnormalities.

New table 5 shows liver stiffness and prevalence of significant fibrosis among four groups (a) and liver stiffness among four group with significant fibrosis.

Because new table 5 included the all information of figure 4, figure 4 was removed.

The information of model 1 and 2, except of model 3 were removed according to the recommendation of reviewer 1, because they had the similar information.

We really appreciated the comments of reviewer 1 and 2, to make our manuscript more informative and appropriately.

We hope our revisions are satisfactory and thank you once again for your consideration of our paper for publication in your journal.

Sincerely,

Dae Won Jun, MD, PhD and Eun-Hee Nah, MD, PhD

Reviewer 1

The manuscript by Dr Park and Yoon et al describes assessment of hepatic fibrosis by magnetic resonance elastography (MRE) in patients from 13 health care centers in Korea. In addition to MRE, patients were screened by abdominal ultrasound for the presence of fatty liver, for metabolic markers by blood test and by routine questionnaires and physical examination. After excluding patients with viral hepatitis and a risk of liver disease, 5111 patients were studied and were assigned to 4 groups: (A) metabolically healthy-no fatty liver (MH-NFL), (B) metabolically unhealthy no fatty liver (MU-NFL), (C) metabolically healthy-fatty liver (MH-FL) and (D) metabolically unhealthy fatty liver (MU-FL). The results of the study are interesting. A total of 7.3% of patients had significant fibrosis and of these 41% were assigned to the MH-NFL and MU-NFL groups indicating that patients with fibrosis might be missed if screening was restricted to patients with fatty liver detected by abdominal ultrasound. The presence of >2 metabolic abnormalities was also an independent risk factor for significant fibrosis.

However, the number of patients with significant or advanced fibrosis is low. Only 1.9% of patients had advanced fibrosis and because of the low patient numbers only limited analysis of patients in this category was provided. The manuscript must be revised to clearly define the number of patients with significant and advanced fibrosis. This could include a new Table 2 showing the number of patients in Groups A-D with significant and advanced fibrosis similar to that shown in the Table section of Figure 4. The number of patients at each level of hepatic stiffness could also be included in Table 4. The M&M section in particular must be revised to eliminate repetitive text.

Comments and suggestions:

  1. Line 2. Consider revising the title to: “Selecting the target population for screening of hepatic fibrosis in primary care centers in Korea.”

☞ Author responses: Thank you for your comment. We changed the title as follows per your suggestions, “Selecting the target population for screening of significant hepatic fibrosis in health check-up program in primary care centers in Korea”. (page 1, line 2-3).

  1. Lines 3, 21, 34, 42-43, 46, 58, 68, 78-81, 87-88. Review use of the terms “health check-up centers”, “health check-up programs”, “health check-up cohorts”, “health promotion centers” and “primary care centers”. Can this be simplified?

☞ Author responses: Thank you for your comment. As your recommendation, we changed “health promotion centers” into health check-up centers.

  1. Lines 6-15. Remove email addresses. There is potential confusion with multiple authors from addresses 2 and 4.

☞ Author responses: Thank you for your comment. As your recommendation, e-mail addresses were removed.

  1. Lines 28 and 29. Review use of the terms metabolic “abnormalities” and metabolic “risks” to see if a consistent descriptor can be used. The words “dysfunction”, “syndrome”, “unhealthy”, “risk abnormalities” and “risks/dysfunction” are all used at other places in the manuscript.

☞ Author responses: Thank you for your comment. Thank you for your comment. As your point-out, it is necessary to use consistent terminology. As shown below, metabolic dysfunction was unified as metabolic abnormality. The terms referring to the components of metabolic risks used in defining metabolic syndrome have been consistently modified to ‘metabolic risk abnormalities’. However, in our study, metabolically unhealthy or healthy status was defined based on our data, so the term was used as it is.

Metabolic dysfunction -> metabolic abnormality

Metabolic risks or metabolic abnormalities -> Metabolic risk abnormalities

  1. Revise to include details of the 1.9% of patients with advanced fibrosis.

☞ Author responses: Thank you for your comment. As your recommendation, new table 2 to show the prevalence of significant and advanced fibrosis of subjects with each components of metabolic risk abnormalities was added.

  1. Lines 21, 24, 82, 89, 91, 101, 105, 106, 126, 227, 270, 328 and Figure 1. Review use of “ultrasonography”, “sonography”, “sonographic” and “ultrasound” to simplify the text.

☞ Author responses: Thank you for your comment. As your point-out, it is necessary to use consistent terminology. We changed these words to “sonography”.

  1. Line 40. Revise to: “defining which patients in the target population need further diagnostic”.

☞ Author responses: Thank you for your comment. Considering the recommendations of two reviewers, we revised as following

“Moreover, subjects with significant hepatic fibrosis have a very high risk of coexisting other gastrointestinal disease (ex. diverticulosis), extrahepatic malignancy and cardiovascular disease beyond liver disease [5,6].”   

  1. Line 44. Revise to: “on the target population who need further”.

☞ Author responses: Thank you for your comment. As your point-out, we revised sentence.

  1. Line 50. Revise to: “Recent published data suggested”.

☞ Author responses: Thank you for your comment. As your recommendation, we revised sentence.

  1. Line 73. Check spelling of “Data”.

☞ Author responses: Thank you for your comment. We made a correction for typo

  1. Lines 65-81. Revise and shorten and consider combining Sections 2.1 and 2.2.

☞ Author responses: Thank you for your comment. As your recommendation, we combined Sections 2.1 and 2.2.

  1. Lines 91-94. Is this information needed?

☞ Author responses: Thank you for your comment. People with chronic liver disease are managed in a specialized program in Korea. Therefore, it is assumed that there is a low chance of including patients with a known risk of hepatic fibrosis in this cohort. We added the above information as an explanation of concern that more people with risk for chronic liver disease may be included in our cohort. If you recommend to remove above information, we can delete the above sentence.

  1. Lines 96-99. Remove duplication with Section 2.2.

☞ Author responses: Thank you for your comment. As your recommendation, we removed sentences with redundancy.

  1. Line 106. Delete “in the MRE and abdominal ultrasound cohorts”

☞ Author responses: Thank you for your comment. As your recommendation, we removed above words.

  1. Figure 1. This Figure can be moved to the Results, Section 3.

☞ Author responses: Thank you for your comment. As your recommendation, we moved figure 1 to result section.

  1. Figure 1. The N=5111 should be n=5111. The details in the RH box “Chronic liver disease” and “Risk for liver disease” are confusing and should be simplified. Can a single heading be used?

☞ Author responses: Thank you for your comment. As your recommendation, we revised figure 1. We added additional information at 2.3. Inclusion and Exclusion Criteria section, as follow “Subjects with risks of chronic liver disease (n=1,665), such as subjects with chronic liver disease on past medical history, positive in viral marker and significant alcohol intake (weekly alcohol consumption >210 g for men and >140 g for women) were excluded.”

  1. Figure 1. Check the box “Non-fatty liver”. Should this be “No fatty liver”?

☞ Author responses: Thank you for your comment and sorry for our mistake. As your point-out, we revised figure 1.

  1. Figure 1. The category “Metabolically unhealthy” applies to Groups B and D and could be placed centrally or included in the footnotes.

☞ Author responses: Thank you for your comment. As your recommendation, we moved the category “Metabolically unhealthy” centrally.

  1. Lines 161-164. Is repetitive with Section 2.4 and Figure 1.

☞ Author responses: Thank you for your comment. As your point-out, we removed the repetitive information.

  1. Table 1. Include the number of female patients in each group. There may be a simpler way to present the P-value data as it doesn’t add much to the Table. The total of the percentages in the row “Characteristics” equals 99.9% not 100%.

☞ Author responses: Thank you for your comment. We added the number of female patients in each group. The percentage of MUFL was rounded to 32.5% (32.4% -> 32.5%).

  1. Lines 167-169. States that the % of patients with significant and advanced fibrosis (7.3% and 1.9%) is presented in Table 1. Can this detail be included in a new Table 2 suggested above?

☞ Author responses: Thank you for your comment. As your recommendation, new table 2 to show the prevalence of significant and advanced fibrosis of subjects with each components of metabolic risk abnormalities was added.

  1. Lines 180-181. States that the prevalence of significant and advanced fibrosis was 12.1% and 3.9% with pre-diabetes or diabetes but the data is not shown. Can this detail be included in a new Table 2 suggested above?

☞ Author responses: Thank you for your comment. As your recommendation, new table 2 to show the prevalence of significant and advanced fibrosis of subjects with each components of metabolic risk abnormalities was added.

Table 2. Prevalence of significant and advanced hepatic fibrosis of subjects with each component of metabolic risk abnormalities

Subjects

Significant fibrosis

Advanced fibrosis

Total subjects n=5,111 (100)

372 (7.3)

95 (1.9)

Subjects with metabolic abnormality

  Central obesity n=1,812 (35.5)

189 (10.4)

48 (2.6)

  High blood pressure or hypertension medication n=1,156 (22.6)

108 (9.3)

29 (2.5)

  High triglyceride n=1,811 (35.6)

170 (9.4)

44 (2.4)

  Low HDL or dyslipidemia medication n=967 (18.9)

87 (9.0)

26 (2.7)

  Prediabetes or diabetes n=1,705 (33.4)

207 (12.1)

67 (3.9)

Data are expressed as number (percent).

Abbreviations: HDL, high-density lipoprotein.

  1. Table 2. Revise heading to “significant hepatic fibrosis”. Do males have a risk factor for fibrosis?

☞ Author responses: Thank you for your comment. As your recommendation, we revised heading to “significant hepatic fibrosis” Male sex was evaluated as risk factor for significant hepatic fibrosis (OR 2.13, P <0.001). Instead of female (negative risk factor), male was added in table 3, because of positive risk factor.

  1. Lines 193-201. Footnotes, Table 2. Revise and shorten and refer to M&M section.

☞ Author responses: Thank you for your comment. As your recommendation, explanation for the definitions of abnormalities in footnotes was removed for simplicity. Moreover, they were described in Method section (2.6. Definition of abnormality).

  1. Figure 2A and 2B. The description of the P-values is confusing.

☞ Author responses: Thank you for your comment. As your recommendation, sentences were revised as follows “* P-value was lower than 0.05, when compared with no fatty liver subjects with 0 metabolic risk abnormality. ** P-value was lower than 0.05, when compared with fatty liver subjects with 0 metabolic risk abnormality.”

  1. Table 3. Models 1, 2 and 3 are not discussed and seem repetitive.

☞ Author responses: Thank you for your comment. In order to increase simplicity and decrease complexity, Model 1 and 2 were removed in table 4. Instead of female sex, male sex was used as variable.

  1. Figure 3. It’s not immediately clear how Figures 3A and 3B relate to 3C. It would be helpful if the colors were matched in Figures 3A and 3B with Figure 3C.
  2. Figure 3C. The vertical axis is labelled “Hepatic fibrosis (%)” but all patients have been selected for “hepatic fibrosis”. Similarly, the heading to Figure 3C does not make sense.
  3. Legend to Figure 3C. Does not mention advanced fibrosis.

☞ Author responses: Thank you for your comment. We agreed with your comment totally. We made graph inappropriately. We are very sorry and very appreciated for important comments. As your point-out, we revised figure 3c. And we revised the manuscript as follow “The proportion of hepatic fibrosis in subjects with metabolically unhealthy status (67.5% in significant, 74.7% in advanced) seemed higher than those in subjects with fatty liver (58.8% in significant, 61.0% in advanced) (Figure 3C).” on page 7, line 224-226. figure legend was also revised.

If this graph results in confusion, it will be good to remove the figure 3C. Because figure 3 A & B also include this information.

  1. Lines 259-261 and Figure 4A. Given the large error bars it is hard to believe there are statistical differences between Groups A-D in the levels of liver stiffness. Can the levels of liver stiffness for the Group A-D patients with significant and advanced fibrosis be provided in a new Table 2?
  2. Figure 4, Table. Consider incorporating the data from the Table into a new Table 2. Figure 4A and 4B if still considered essential could be incorporated into Figure 3.

☞ Author responses: Thank you for your comment. As your recommendation, we added new table 5 to including the information on the liver stiffness value of four group with significant fibrosis. However, the number of subjects with advanced fibrosis was so small, our study focused on the subjects with significant fibrosis, new table 5 did not include the information on liver stiffness value of four group with advanced fibrosis. Because new table 5 including the all information of figure 4, figure 4 was remove.

This information was added as follow

“Although the statistical comparison of liver stiffness value among four groups with significant fibrosis, due to the small number of subjects with significant fibrosis, the liver stiffness values of metabolic unhealthy group such as MU-NFL (3.65±1.15 kPa), and MU-FL (3.57±0.79 kPa) were statistically higher than that of MH-NFL (3.35±0.41 kPa).”

Table 5. Liver stiffness and prevalence of significant fibrosis among four groups (a) and liver stiffness among four group with significant fibrosis.

A) MH-NFL

B) MU-NFL

C) MH-FL

D) MU-FL

P-value

A vs B

A vs C

A vs D

B vs C

B vs D

C vs D

a) Four group

Liver stiffness (kPa)†

2.24±0.44

2.33±0.64

2.27±0.42

2.43±0.58

<0.001

0.044

<0.001

0.05

<0.001

<0.001

Significant fibrosis

85 (4.5)

68 (9.2)

36 (4.4)

183 (11)

<0.001

0.868

<0.001

<0.001

0.166

<0.001

b) Four group with significant fibrosis

Liver stiffness (kPa)†

3.35±0.41

3.65±1.15

3.35±0.79

3.57±0.79

0.028

0.938

0.015

0.141

0.550

0.107

Data are expressed as number (percent). † Data are shown as mean ± standard deviation. Abbreviations: MH-NFL, metabolically healthy, non-fatty liver; MU-NFL, metabolically unhealthy, non-fatty liver; MH-FL, metabolically healthy, fatty liver; MU-FL, metabolically healthy, fatty liver.

  1. Table 4. It would be useful to add the number of patients at each level of hepatic stiffness (>3 kPa – >4.2kPa) to help support the data.

☞ Author responses: Thank you for your comment. As your recommendation, the number of patients at each level of hepatic stiffness was added in Table 5.

  1. Reference 2. Check page numbers.

☞ Author responses: Thank you for your comment. We confirmed the citation information again. This reference don’t have any page number.

  1. Reference 4. Check volume and year.

☞ Author responses: Thank you for your comment. We revised the information of reference.

  1. Reference 8. Check page numbers.

☞ Author responses: Thank you for your comment. We revised the information of reference.

  1. Reference 10. Check volume and page numbers.

☞ Author responses: Thank you for your comment. We revised the information of reference.

  1. Reference 13. Check volume.

☞ Author responses: Thank you for your comment. We revised the information of reference.

Reviewer 2 Report

This is a well executed study about the presence of liver fibrosis is patients with and without components of metabolic syndrome. The study sample is large, above 5000. Authors fine that the prevalence of fibrosis is around 7% but of these patients 60% had fatty liver, which brings interesting dilemma- what is pathophysiology of the fibrosis in reminder of the patients. Patients with viral hepatitis and alcohol use were excluded. 

  1. Title: "at risk" should be added in front of "for". Manuscript should be checked thoroughly for grammar
  2. Introduction- I would appreciate more informative introduction. For example,  more information regarding conditions associated with NAFLD such as diverticulosis should be mentioned ( https://pubmed.ncbi.nlm.nih.gov/33951119/) . Consequently, patients diagnosed with diverticulosis on screening colonoscopy might be a good candidate for liver fibrosis screening.
  3. Discussion- co existence of fatty pancreas and NAFLD and their association with component of metabolic syndrome should be discussed.
  4. What is the rationale for not checking HgbA1c level or insulin level? Insulin resistance is the cornerstone of fatty liver and vice versa.
  5.  How do you explain that 41% of subjects had fibrosis without fatty liver? 

Author Response

Dear Reviewers and editors in JCM

Thank you very much for your time and consideration of our manuscript. We appreciate the opportunity to respond to the comments by the reviewers and to improve our paper. Please find below our responses to reviewers’ comments.

If we summarized the main change in Figure and tables.

HbA1c data was added in table 1, according to the recommendation of reviewer 2.

New table 2 and table 5 were added according to the recommendation of reviewer 1.

New table 2 shows prevalence of significant and advanced hepatic fibrosis of subjects with each component of metabolic risk abnormalities.

New table 5 shows liver stiffness and prevalence of significant fibrosis among four groups (a) and liver stiffness among four group with significant fibrosis.

Because new table 5 included the all information of figure 4, figure 4 was removed.

The information of model 1 and 2, except of model 3 were removed according to the recommendation of reviewer 1, because they had the similar information.

We really appreciated the comments of reviewer 1 and 2, to make our manuscript more informative and appropriately.

We hope our revisions are satisfactory and thank you once again for your consideration of our paper for publication in your journal.

Sincerely,

Dae Won Jun, MD, PhD and Eun-Hee Nah, MD, PhD

Reviewer 2

This is a well executed study about the presence of liver fibrosis is patients with and without components of metabolic syndrome. The study sample is large, above 5000. Authors fine that the prevalence of fibrosis is around 7% but of these patients 60% had fatty liver, which brings interesting dilemma- what is pathophysiology of the fibrosis in reminder of the patients. Patients with viral hepatitis and alcohol use were excluded. 

  1. Title: "at risk" should be added in front of "for". Manuscript should be checked thoroughly for grammar

☞ Author responses: Thank you for your comment. We changed the title as follows, “Selecting the target population for screening of hepatic fibrosis in in primary care centers in Korea”. on page 1, line 2-3.

  1. Introduction- I would appreciate more informative introduction. For example, more information regarding conditions associated with NAFLD such as diverticulosis should be mentioned ( https://pubmed.ncbi.nlm.nih.gov/33951119/) . Consequently, patients diagnosed with diverticulosis on screening colonoscopy might be a good candidate for liver fibrosis screening.

Author responses: Thank you for your comment. We think that your comment is important issue. However, our data is not enough to support above opinion “Patients diagnosed with diverticulosis on screening colonoscopy might be a good candidate for liver fibrosis screening.”

Nevertheless, we think that introduction of your comments make our induction more informative. So we revised our introduction slightly, as following

“Considering the direct relationship between significant hepatic fibrosis and the risk of liver-related mortality [3,4], screening strategy and defining high risk group for significant hepatic fibrosis among average risk group are important in primary care centers. Moreover, subjects with significant hepatic fibrosis have a very high risk of coexisting other gastrointestinal disease (ex. diverticulosis), extrahepatic malignancy and cardiovascular disease beyond liver disease [5,6].”

  1. Discussion- co existence of fatty pancreas and NAFLD and their association with component of metabolic syndrome should be discussed.

Author responses: Thank you for your comment. As your recommendation, we added below paragraph and reference.

“Recently, several studies suggested that non-alcoholic fatty pancreas disease (NAFPD) might be use as surrogate marker for pre-diabetics or metabolic dysfunction [24]. Further studies needed on clinical implication of co-existence of NAFPD and NAFLD.

  1. What is the rationale for not checking HgbA1c level or insulin level? Insulin resistance is the cornerstone of fatty liver and vice versa.

☞ Author responses: Thank you for your comment. We totally agreed with your opinion that Insulin resistance is the cornerstone of fatty liver. HbA1c level was included as the component of metabolic risk abnormality, when to evaluate the IFG of subjects. However, insulin level or HOMA-IR could not be included in our study, because insulin level was nor evaluated in health check-up program generally. Due to the nature of our cohort, prevalence of subjects with hepatic fibrosis is very low. For getting statistical significance, we cannot help including subjects without the data of serum insulin level.

We added this information as our limitation as following

“Moreover, important metabolic risk abnormalities such as serum insulin level, and CRP could not be included in this analysis, because they were not examined routinely in health check-up program.”  

  1. How do you explain that 41% of subjects had fibrosis without fatty liver?

☞ Author responses: Thank you for your comment. We think that other risk factors, such as past hepatitis B infection, genetic differences or single nucleotide polymorphisms, intestinal microbial composition, and sarcopenia, could result in hepatic fibrosis.

Moreover, the limitation of sonographic evaluation could result in misdiagnosis, because the sensitivity of sonography to detect the presence of fatty liver is largely dependent on the skill and expertise of the examiners. However, most subjects with fatty liver are diagnosed with the abdominal sonography in real-life settings. Therefore, it is necessary that screening for hepatic fibrosis is extended to subjects with metabolically unhealthy status, beyond those with fatty liver, in order to reduce the risk of missing subjects with significant fibrosis.

This information was included in discussion section (page 9-10, line 290-294 and Page 10 line 298-308).

Round 2

Reviewer 1 Report

The manuscript by Dr Park and Yoon et al has been improved by revision. However, further changes are required as outlined below. Care must be taken to ensure that the description of the data matches the sequence and layout of the figures and tables. Tables 3 and 4 can be combined as Table 3A and 3B. Table 5 can be omitted. Table 6 should be reviewed and discussed or omitted.

  1. Line 3. Check the revised title and remove “in” or insert “health check-up programs”.
  2. Line 23. Revise to “The prevalence of significant and advanced hepatic fibrosis was 7.3% and 1.9% respectively.”
  3. Line 40. Check spelling of “diverticulitis”.
  4. Line 67-68. Delete “between January 2017 and May 2020”. The dates are provided on line 99.
  5. Line 77-78. Revise to “that have MRE facilities installed were selected for the current study.”
  6. Line 105. Heading 2.4. Add italics and remove capital from “parameters”.
  7. Line 111. Define BMI.
  8. Section 2.4 and Table 1. Define HbA1c in the text and footnotes to Table 1. Provide details of the normal range of hemoglobin A1c in %.
  9. Line 140 and Table 3 and 4. “Abnormal aminotransferase” is identified as a risk factor in both Table 3 and Table 4. Is it also a “metabolic risk abnormality”? If so the definition of “abnormal aminotransferase” should be moved from line 140 to line 132.
  10. Lines 137-139. The 4 groups should be labelled A-D (not 1-4).
  11. Figure 1 and lines 137-139, 162-163, 167-168, 253-254. Groups A and B should be named consistently. Group A = MH-NFL = metabolically healthy, non-fatty liver and Group B = MU-NFL = metabolically unhealthy, non-fatty Note that the description is different in lines 137-139 and in 3 boxes in Figure 1.
  12. Lines 158-160. The sentence starting “All subjects” should be moved to line 155 and referenced to Figure 1 and Table 1.
  13. Line 158. Check % and revise to “Furthermore, 14.5% of patients with non-fatty liver were metabolically unhealthy”.
  14. Lines 155-156. Revise the sentence and move to line 172 at the beginning of Section 3.2. “The proportion of patients with significant and advanced fibrosis was 7.3 and 1.9% respectively (Table 2).”
  15. Lines 166-170. List the footnotes in the order that they appear in the table.
  16. Section 3.2 and Table 2. The data in Table 2 must be discussed in the text. Description of the data is currently lacking. Section 3.2 could begin with a statement like “Within the total of the 5111 patients, 372 (7.3%) had significant fibrosis (MRE score 3.0-3.4 kPa) and 95 (1.9%) had advanced fibrosis (MRE score 3.6-4.2 kPa) (Table 2). Patients were identified with metabolic abnormalities that included…”
  17. Table 2. Footnotes need to be added to explain the percentage values shown in brackets. For example: Central obesity n=1812 (35.5) >> 189 (10.4) >> 48 (2.6). The 35.5% is the percentage of the 5111 total patients with central obesity. The 10.4% is the percentage of the 1812 patients with central obesity that have significant fibrosis etc.
  18. Lines 179 and 181. Replace “abnormal” with “high” to match Tables 2 and 3.
  19. Section 3.2 and 3.3 and Table 3. Similarly, the data in Table 3 must be discussed in the text. Table 3 is only mentioned once in Section 3.2 and once in Section 3.3.
  20. Footnotes Table 3. “LFT, liver function test” can be deleted as it is not shown in the Table and is not used in the text.
  21. Combine Table 3 and 4. Can Table 3 be presented as 3A and Table 4 be presented as 3B?
  22. Figure 3. The color coding makes the Figure much easier to understand.
  23. Figure 3A and 3B. Add the patient numbers in each Group A-D to the labels on the pie charts. The patient numbers are currently shown in Table 5. In Figure 3A the black slice would be labelled n=85 (22.8%), the red slice n=68 (18.3%), the green slice n=36 (9.7%) and blue slice n=183 (49.2%) etc.
  24. Delete the heading to Figures 3A, 3B and 3C.
  25. Legend to Figure 3. Instead of “four groups” use “Groups A-D”. The legend does not make sense as all patients have fibrosis. The legend can be revised to: “Venn diagram representing the proportion of patients in Groups A-D with (A) significant (n=372) and (B) advanced fibrosis (n=95). (C) The proportion of patients with fatty liver and metabolically unhealthy status with significant and advanced fibrosis.”
  26. Figure 3C. The vertical axis does not make sense as all patients have fibrosis and should be revised to: “Proportion of patients (%)”.
  27. Section 3.4 and Figure 3. The description of the data in the text is cryptic and should be revised. This is the most important Figure in the manuscript and it must be described carefully.
  28. Lines 225-230 and Table 5. The discussion of the liver stiffness data can be omitted along with Table 5.
  29. Table 6. The prevalence data does not make sense as the number of patients with metabolically unhealthy state and fatty liver cannot be the same and the numbers do not add up to n=372 and n=95. Table 6 should be carefully revised and discussed or omitted.

Author Response

The manuscript by Dr Park and Yoon et al has been improved by revision. However, further changes are required as outlined below. Care must be taken to ensure that the description of the data matches the sequence and layout of the figures and tables. Tables 3 and 4 can be combined as Table 3A and 3B. Table 5 can be omitted. Table 6 should be reviewed and discussed or omitted.

  1. Line 3. Check the revised title and remove “in” or insert “health check-up programs”.

☞ Thank you for your comment and we are really sorry for the typo. We removed duplicated “in” from the title.

  1. Line 23. Revise to “The prevalence of significant and advanced hepatic fibrosis was 7.3% and 1.9% respectively.”

☞ Thank you for your comment. We added the prevalence of advanced hepatic fibrosis as follows, “The prevalence of significant and advanced hepatic fibrosis was 7.3% and 1.9% respectively.”

  1. Line 40. Check spelling of “diverticulitis”.

☞ Thank you for your comment. We rechecked the spelling.

  1. Line 67-68. Delete “between January 2017 and May 2020”. The dates are provided on line 99.

☞ Thank you for your comment. We removed the dates for the readability.

  1. Line 77-78. Revise to “that have MRE facilities installed were selected for the current study.”

☞ Thank you for your comment. As your recommendation, sentence was revised.

  1. Line 105. Heading 2.4. Add italics and remove capital from “parameters”.

☞ Thank you for your meticulous advise. We made a correction.

  1. Line 111. Define BMI.

☞ Thank you for your comment. As your recommendation, “body mass index (BMI, weight/height2)” was added in line 112-114

  1. Section 2.4 and Table 1. Define HbA1c in the text and footnotes to Table 1. Provide details of the normal range of hemoglobin A1c in %.

☞  : Thank you for your comment. As your recommendation, “hemoglobin A1c (HbA1c)” in section 2.4 and “HbA1c, hemoglobin A1c” in table 1 was added, respectively. The normal range of hemoglobin A1c in % was added in section 2.6 (line133-135), as follows

“5) prediabetes or diabetes, fasting glucose level ≥100 mg/dL, HbA1c ≥ 5.7%, and/or taking diabetes medication.”

  1. Line 140 and Table 3 and 4. “Abnormal aminotransferase” is identified as a risk factor in both Table 3 and Table 4. Is it also a “metabolic risk abnormality”? If so the definition of “abnormal aminotransferase” should be moved from line 140 to line 132.

☞  : Thank you for your comment. Abnormal aminotransferase is not routinely considered as the metabolic risk abnormality. However, it is well known risk factor for hepatic fibrosis. Therefore, it was included as a covariate in the multivariable analysis (table 3 and 4)

  1. Lines 137-139. The 4 groups should be labelled A-D (not 1-4).

☞ Thank you for your comment. As your recommendation, 4 groups were labelled A-D.

  1. Figure 1 and lines 137-139, 162-163, 167-168, 253-254. Groups A and B should be named consistently. Group A = MH-NFL = metabolically healthy, non-fatty liver and Group B = MU-NFL = metabolically unhealthy, non-fatty Note that the description is different in lines 137-139 and in 3 boxes in Figure 1.

☞ Thank you for your comment. We revised to use the same terminology in Figure 1, Table 1(line 168), and Figure 2 (boxes).

  1. Lines 158-160. The sentence starting “All subjects” should be moved to line 155 and referenced to Figure 1 and Table 1.

☞ Thank you for your comment. As your recommendation, sentence was revised as follow

 “All subjects were divided into four groups according to the presence of fatty liver and metabolic health status (Figure 1 and table 1).” (line 157-159)  

  1. Line 158. Check % and revise to “Furthermore, 14.5% of patients with non-fatty liver were metabolically unhealthy”.

☞ Thank you for your comment. As your recommendation, sentence was revised as follow

“Furthermore, 28% of patients with non-fatty liver (742/2,628) were metabolically unhealthy.” (line 160-161). For decreasing confusion, we added the number.

  1. Lines 155-156. Revise the sentence and move to line 172 at the beginning of Section 3.2. “The proportion of patients with significant and advanced fibrosis was 7.3 and 1.9% respectively (Table 2).”

☞ Thank you for your comment. As your recommendation, sentence was moved to line 174

  1. Lines 166-170. List the footnotes in the order that they appear in the table

☞  : Thank you for your comment. As your recommendation, the footnotes was placed in the order that they appear in the table.

  1. Section 3.2 and Table 2. The data in Table 2 must be discussed in the text. Description of the data is currently lacking. Section 3.2 could begin with a statement like “Within the total of the 5111 patients, 372 (7.3%) had significant fibrosis (MRE score 3.0-3.4 kPa) and 95 (1.9%) had advanced fibrosis (MRE score 3.6-4.2 kPa) (Table 2). Patients were identified with metabolic abnormalities that included…”

☞  : Thank you for your comment. As your recommendation, additional explanation was included as follow

“Within the total of the 5111 patients, 372 (7.3%) had significant fibrosis (MRE score ≥ 3.0 kPa) and 95 (1.9%) had advanced fibrosis (MRE score ≥ 3.6 kPa) (Table 2). The prevalence of advanced hepatic fibrosis two times higher in prediabetes or diabetes subjects compare to total population (1.9% vs. 3.9%). Prevalence of significant and advanced hepatic fibrosis was 10.4% and 2.6%, respectively in case of central obesity. Overall prevalence of subjects with central obesity, high blood pressure or hyper-tension medication, high triglyceride level, low HDL or dyslipidemia medication, and prediabetes or diabetes in total subjects were 33.5, 22.6, 35.6, 18.9, and 33.4%, respectively.” (line 175-182)

  1. Table 2. Footnotes need to be added to explain the percentage values shown in brackets. For example: Central obesity n=1812 (35.5) >> 189 (10.4) >> 48 (2.6). The 35.5% is the percentage of the 5111 total patients with central obesity. The 10.4% is the percentage of the 1812 patients with central obesity that have significant fibrosis etc.

☞ Thank you for your comment. As your recommendation, footnote was added as follow

“*Prevalence of significant or advanced hepatic fibrosis indicates the proportion of subjects with significant or advanced hepatic fibrosis among total subjects (n-5,111), subjects with central obesity (n=1,812), high blood pressure or hypertension medication (n=1,156), high triglyceride (n=1,811), low HDL or dyslipidemia medication (n=967), or prediabetes or diabetes (n=1,705).” (line 189-192)

  1. Lines 179 and 181. Replace “abnormal” with “high” to match Tables 2 and 3.

☞ Thank you for your comment. As your recommendation, “abnormal” was changed to “high” (line 181).

  1. Section 3.2 and 3.3 and Table 3. Similarly, the data in Table 3 must be discussed in the text. Table 3 is only mentioned once in Section 3.2 and once in Section 3.3.

☞ Thank you for your comment. As your recommendation, the explanation about table 3 was added as follow

“In order to evaluate the independent risk factor for significant hepatic fibrosis, univariate and multivariate analyses were done (Table 3). Age, Male sex, BMI, and the presence of fatty liver, hypertension, diabetes, abnormal aminotransferase, and all five components of metabolic risk abnormalities in metabolic syndrome were associated with a higher risk of significant fibrosis in univariate analysis (Table 3a). Metabolically unhealthy status was evaluated as significant predictors in not only univariate but also multivariate analysis (OR 1.76, 95% CI 1.37-2.26, P<0.001) (Table 3b), In contrast, the presence of fatty liver was not an independent risk factor for significant fibrosis.” (Line 211-218)

  1. Footnotes Table 3. “LFT, liver function test” can be deleted as it is not shown in the Table and is not used in the text.

☞ Thank you for your comment. As your recommendation, “LFT, liver function test” was removed.

  1. Combine Table 3 and 4. Can Table 3 be presented as 3A and Table 4 be presented as 3B?

☞ Thank you for your comment. As your recommendation, Table 3 and 4 were combined.

  1. Figure 3. The color coding makes the Figure much easier to understand.

☞ Thank you. Your comment makes figure 3 more appropriately.

  1. Figure 3A and 3B. Add the patient numbers in each Group A-D to the labels on the pie charts. The patient numbers are currently shown in Table 5. In Figure 3A the black slice would be labelled n=85 (22.8%), the red slice n=68 (18.3%), the green slice n=36 (9.7%) and blue slice n=183 (49.2%) etc.

☞ Thank you for your comment. As your recommendation, the patient numbers were added.

  1. Delete the heading to Figures 3A, 3B and 3C.

☞ Thank you for your comment. As your recommendation, the heading to Figures 3A, 3B and 3C were removed.

  1. Legend to Figure 3. Instead of “four groups” use “Groups A-D”. The legend does not make sense as all patients have fibrosis. The legend can be revised to: “Venn diagram representing the proportion of patients in Groups A-D with (A) significant (n=372) and (B) advanced fibrosis (n=95). (C) The proportion of patients with fatty liver and metabolically unhealthy status with significant and advanced fibrosis.”

☞ Thank you for your comment. As your recommendation, Legend to Figure 3 was revised.

  1. Figure 3C. The vertical axis does not make sense as all patients have fibrosis and should be revised to: “Proportion of patients (%)”.

☞ Thank you for your comment. As your recommendation, the vertical axis was modified.

  1. Section 3.4 and Figure 3. The description of the data in the text is cryptic and should be revised. This is the most important Figure in the manuscript and it must be described carefully.

☞ Thank you for your comment. As your recommendation, further explanation was added as follow

“A total of 41.1% and 39.0% of those with significant and advanced fibrosis did not combine fatty liver (black & red pie) (Figures 3A and 3B). A total 67.5% and 74.7% of subjects with significant fibrosis and advanced hepatic fibrosis were metabolically unhealthy (red & blue pie). The proportion of subjects with only fatty liver among those with significant and advanced fibrosis (green pie) were 9.7% and 8.4%, respectively. The proportion of subjects with only metabolically unhealthy status among those with significant and advanced fibrosis (red pie) were 18.3% and 22.1%, respectively.” Line 227-233

  1. Lines 225-23 0 and Table 5. The discussion of the liver stiffness data can be omitted along with Table 5.

☞ Thank you for your comment. As your recommendation, the data and discussion about the liver stiffness of four groups with significant fibrosis were removed.

  1. Table 6. The prevalence data does not make sense as the number of patients with metabolically unhealthy state and fatty liver cannot be the same and the numbers do not add up to n=372 and n=95. Table 6 should be carefully revised and discussed or omitted.

☞ Thank you for your comment. We think that table 6 is not essential component in our manuscript and has the possibility to result in confusion. Therefore, we would like to omit table 6.
